# Worry about Radiation and Its Risk Factors Five to Ten Years after the Fukushima Nuclear Power Plant Disaster

**DOI:** 10.3390/ijerph192416943

**Published:** 2022-12-16

**Authors:** Maiko Fukasawa, Maki Umeda, Tsuyoshi Akiyama, Naoko Horikoshi, Seiji Yasumura, Hirooki Yabe, Yuriko Suzuki, Evelyn J. Bromet, Norito Kawakami

**Affiliations:** 1Health Promotion Center, Fukushima Medical University, Fukushima 960-1295, Japan; 2Research Institute of Nursing Care for People and Community, University of Hyogo, Akashi 673-8588, Japan; 3Department of Neuropsychiatry, NTT Medical Center Tokyo, Shinagawa-ku, Tokyo 141-8625, Japan; 4Radiation Medical Science Center for the Fukushima Health Management Survey, Fukushima Medical University, Fukushima 960-1295, Japan; 5Department of Public Health, Fukushima Medical University School of Medicine, Fukushima 960-1295, Japan; 6Department of Neuropsychiatry, Fukushima Medical University School of Medicine, Fukushima 960-1295, Japan; 7Department of Public Mental Health Research, National Institute of Mental Health, National Center of Neurology and Psychiatry, Kodaira 187-8551, Japan; 8Department of Psychiatry and Behavioural Health, Stony Brook University, Stony Brook, NY 11794, USA; 9Department of Digital Mental Health, Graduate School of Medicine, The University of Tokyo, Bunkyo-ku, Tokyo 113-0033, Japan

**Keywords:** Fukushima, nuclear power plant accident, worry about radiation, disaster-related experiences, longitudinal study

## Abstract

Worry about radiation persists long after nuclear power plant accidents. Young age, low socioeconomic status, being married, and disaster-related experiences are known to be associated with greater worry about radiation. This study explored the duration of the effects of these risk factors on worry about radiation after the 2011 Fukushima nuclear power plant accident, using the longitudinal data of randomly sampled non-evacuee community residents who were followed five to ten years after the accident. Questionnaire surveys were conducted five times with 1825 respondents (37.2% of the 4900 initial targets). We examined the interaction of time and risk factors of worry about radiation using a mixed model. Fear or anxiety immediately after the accident had effects on worry about radiation that continued even after 10 years, though it slightly attenuated with time. Family problems stemming from the disaster retained their effects. While direct damage and evacuation experience were significantly associated with worry about radiation in the early phase, their effects diminished and became non-significant during the study period. Being under the age of 65, having low educational attainment, and being married were significantly associated with worry about radiation, although the association with age weakened over time. Individuals who experience intense fear or anxiety post-nuclear power plant accidents or disaster-related family problems may need continuous monitoring for their worry about radiation even 10 years after such accidents.

## 1. Introduction

Worry about radiation among community residents who experience nuclear power plant accidents persists over the long term, exerting deleterious effects on their mental health [1,2,3,4,5,6,7,8,9,10]. A study after the Chernobyl accident reported that evacuees’ risk perceptions of the accident’s adverse effects on their lives continued to affect their mental health even 19 years after the accident [1].

After the 2011 Fukushima nuclear power plant accident, greater worry about radiation was found to be associated with young age, having low educational attainment, low income, being married, and disaster-related experiences [5,9,11,12]. Regarding age, a study assessing evacuees’ three-year risk perception trajectory for the delayed and genetic effects of radiation exposure reported that those aged 15–49 years had higher risk perception compared to those aged 50–64 years [12]. A study assessing non-evacuees’ radiation anxiety five years after the accident reported that those aged 20–64 years had higher anxiety compared to those aged 65 years and above [5]. Regarding educational attainment, a study assessing evacuees’ risk perception one year after the accident reported that individuals educated up to high school had higher risk perception compared to those with higher educational attainment [9]. A study assessing evacuees’ three-year risk perception trajectory reported that individuals who were educated up to junior high school had higher risk perception and those who graduated from college or higher had lower risk perception compared to those who graduated high school [12]. A study assessing non-evacuees’ radiation anxiety reported that individuals educated up to high school had higher radiation anxiety compared to those who graduated from college or higher [5]. Regarding income, a study assessing non-evacuees’ radiation anxiety reported that those with low and mid-level household income had higher radiation anxiety compared to those with higher household income [5]. Regarding marital status, a study assessing community residents’ perception of radiation risk five years after the accident reported that those with spouses had higher risk perception compared to those without [11]. A study assessing non-evacuees’ radiation anxiety reported that married individuals had higher radiation anxiety compared to those who were separated, divorced, widowed, unmarried, or with an unknown status [5]. Since the Fukushima nuclear power plant accident occurred following the Great East Japan Earthquake, disaster-related experiences included damage caused by not only the nuclear power plant accident but also the massive earthquake and tsunami. Intense fear after the nuclear power plant accident, witnessing the hydrogen plant explosion, damage to houses, injury, bereavement, family separation, evacuation, and loss of job were reported to be associated with greater worry about radiation after the Fukushima nuclear power plant accident [5,9,11,12].

Although worry about radiation remained among community residents in Fukushima even 10 years after the nuclear power plant accident, it gradually decreased both among evacuees and non-evacuees [13,14,15]. However, it is unknown whether the effects of the above-mentioned risk factors, which were reported one-to-five years after the accident [5,9,11,12], persisted for 10 years. It is possible that some factors had retained strong effects while the effects of others had waned with time. Identifying the risk factors with long-lasting effects on worry about radiation will be helpful to not only monitor the affected residents of Fukushima over the long term but also prepare appropriate responses to future nuclear power plant accidents.

Therefore, in this study, we explored the time course of the effects of risk factors on worry about radiation using longitudinal data obtained five to ten years after the nuclear power plant accident from the non-evacuee community residents in Fukushima. We examined the effects of disaster-related experiences as well as sociodemographic characteristics such as age, educational attainment, household income, and marital status, which were consistently reported to be associated with worry about radiation in Fukushima [5,9,11,12].

## 2. Materials and Methods

### 2.1. Study Design and Population

In this study, we used data obtained from a longitudinal survey conducted during a period five to ten years after the 2011 Fukushima nuclear power plant accident. The survey was administered in 49 municipalities of Fukushima Prefecture, excluding the restricted areas designated by the Japanese government close to the nuclear power plant. Of these 49 municipalities, 4 were located in the eastern coastal area (Hama-dori), 28 were in the central area (Naka-dori), and 17 were in the western mountainous area (Aizu). From each municipality, 100 residents aged 20–80 years were randomly sampled, with double weighting for residents aged 20–39 years. This yielded a total of 4900 initial subjects. During the five-year study period, questionnaire surveys were conducted five times through mail. The baseline survey was conducted between February and April 2016. The second, third, fourth, and fifth surveys were conducted 20, 32, 44, and 56 months after the baseline survey, respectively, the targets of which were respondents to the preceding survey [5,6,16]. We used the same questionnaire for all five surveys, with the addition of several pieces of basic information (e.g., disaster-related experiences) for the baseline survey. The number of questionnaires distributed and returned (response rate) in each survey were 4900 and 2038 (41.6%), 2037 and 1450 (71.2%), 1129 and 1013 (89.7%), 927 and 860 (92.8%), and 813 and 774 (95.2%), respectively.

### 2.2. Study Variables

#### 2.2.1. Worry about Radiation

We defined worry about radiation as a negative sense of cognition and perception, such as anxiety about the possible adverse health effects of radiation exposure, and certain related psychosocial problems, such as perceived stigma and discrimination due to radiation exposure [5]. In this study, we assessed worry about radiation using the Radiation Anxiety Scale developed by Umeda et al. [5,17,18]. Its seven items were rated on a 4-point Likert scale ranging from 1 (strongly disagree) to 4 (strongly agree), and their scores were summed to obtain a total score, ranging from 7 to 28; higher scores indicated greater worry about radiation.

#### 2.2.2. Sociodemographic Characteristics

The sociodemographic characteristics included in this study were age, sex, educational attainment, household income in the past year, marital status, and residential area assessed at the baseline survey. The residential area was divided into eastern coastal (Hama-dori), central (Naka-dori), and western (Aizu) areas.

#### 2.2.3. Disaster-Related Experiences

We sought information from respondents regarding four experiences related to the Great East Japan Earthquake in the baseline survey:Fear or anxiety immediately after the nuclear power plant accident;Direct damage caused by the disaster;Disaster-related family problems;Evacuation to avoid radiation exposure.

Fear or anxiety immediately after the nuclear power plant accident was assessed using a 5-point scale ranging from 1 (not at all) to 5 (extremely), and the respondents were dichotomized into those who experienced intense fear or anxiety (scored 5) and others (scored 1–4).

With regard to direct damage caused by the Great East Japan Earthquake, we sought information about four experiences: (a) self-injury, (b) injury or death of family members, (c) loss of job or temporary absence from work, and (d) loss of house or property.

For disaster-related family problems, we sought information about two experiences: (a) deterioration of family relationships, and (b) family separation.

As for the experience of evacuation to avoid radiation exposure after the nuclear power plant accident, we categorized respondents who evacuated themselves as well as those who evacuated only their family but did not evacuate themselves into the evacuated group and compared them with those who did not evacuate at all.

### 2.3. Statistical Analysis

First, we calculated descriptive statistics for sociodemographic characteristics and disaster-related experiences of the study participants assessed at the baseline survey (Time 1) and their worry about radiation during the study period (Time 1 to Time 5). Since the number of respondents decreased over the study period, we compared the level of respondents’ worry about radiation with that of those who dropped out at the subsequent survey. We also calculated the score of worry about radiation only among those who responded to all five time surveys.

Second, we explored the effects of time, sociodemographic characteristics, and disaster-related experiences on worry about radiation, using mixed model repeated measures. After confirming their bivariate associations, we examined the interaction of time with age, educational attainment, income, marital status, and each disaster-related experience, while controlling for sex and residential area using multilevel multivariate linear regression analyses. Regarding sociodemographic characteristics, we explored worry about radiation in those younger than 65, those whose educational attainment was up to high school, those with low or middle level household income, and those married, as these populations showed greater worry about radiation five years after the nuclear power plant accident [5].

All statistical analyses were performed using Stata 17 for Windows (StataCorp LP, College Station, TX, USA). Statistical significance was set at 0.05 and all tests were two-tailed.

## 3. Results

Of the 2038 respondents of the baseline survey (41.6% of the initial target), we included 1825 in the current study, as they had no missing information on sociodemographic characteristics and disaster-related experiences in the baseline survey and had a score of worry about radiation in at least one survey (89.5% of the respondents to the baseline survey and 37.2% of the initial targets). The response rates in the initial targets were significantly different by residential area, namely 34.5%, 36.0%, and 39.9% in the eastern coastal, central, and western mountainous areas, respectively (chi2 = 8.1, df = 2, *p* = 0.017).

Table 1 reports the sociodemographic characteristics of our study participants at Time 1 and their disaster-related experiences in the Great East Japan Earthquake. Individuals who graduated from junior or technical college or a higher educational institution comprised 38.3% of the respondents and those who were married accounted for 63.5%. Individuals who felt intense fear or anxiety immediately after the nuclear power plant accident comprised 36.1%. Those who had been injured, had family members who had been injured or died, had lost jobs or experienced a temporary absence from work, or had lost a house or property in the Great East Japan Earthquake accounted for 31.4%. Individuals who experienced deterioration of their family relationships or were living apart from their family due to the disaster accounted for 7.9%. Individuals who evacuated to avoid radiation exposure after the accident comprised 14.7%.

Figure 1 shows the time course change of worry about radiation during the study period. The Radiation Anxiety Scale score at Time 1, five years after the nuclear power plant accident, was 14.9, which decreased to 13.5 at Time 5, 10 years after the accident. Since the number of respondents decreased for each survey, we compared the score between those who responded to the next survey and those who dropped out at every survey, which revealed no statistically significant differences between them (Appendix A). The time course change of worry about radiation only among those who responded to all five time surveys (*n* = 607) showed similar results (Appendix A).

Table 2 reports the associations of time since the baseline survey, sociodemographic characteristics, and disaster-related experiences with worry about radiation explored using mixed model repeated measures. Concerning bivariate associations, time was significantly and negatively associated with worry about radiation. Being under the age of 65, being married, and living in central or eastern coastal areas were associated with greater worry about radiation. Disaster-related experiences including fear or anxiety immediately after the accident, direct damage, disaster-related family problems, and evacuation experience were all significantly and positively associated with worry about radiation. In Model 1, considering time, sociodemographic characteristics, and disaster-related experience simultaneously, except for household income and evacuation experience, all of the factors were significantly associated with worry about radiation. In Model 2, adding the interactional terms of time with sociodemographic characteristics and disaster-related experiences, the interaction of time was found to be significant with age, fear or anxiety immediately after the accident, direct damage, and evacuation experience. 

To scrutinize the effects of each risk factor and their interaction with time, we plotted the predicted means of worry about radiation of those with and without each risk factor at each time point (Figure 2 and Figure 3). Figure 2 shows the effects of sociodemographic characteristics and their interaction with time on worry about radiation. With regard to age, individuals aged 20–64 years had greater worry about radiation at the baseline survey compared to those aged 65 years or above. However, worry about radiation among those aged 20–64 years declined faster, with the 95% confidence intervals of adjusted prediction of worry about radiation of these two age groups overlapping after Time 2 (i.e., 20 months after the baseline survey). Educational attainment and marital status were associated with worry about radiation and did not interact with the time. The 95% confidence intervals of adjusted predictions of worry about radiation among those with low educational attainment and those who were married at each time point almost overlapped those of their counterparts. Household income was not significantly associated with worry about radiation.

Figure 3 shows the effects of disaster-related experiences and their interaction with time on worry about radiation. Concerning fear or anxiety immediately after the accident, the decline in the level of worry about radiation was faster among those who felt intense fear or anxiety immediately after the accident compared to those who did not experience intense fear or anxiety, and the discrepancy in the level of worry about radiation between these two groups diminished with time. However, those with intense fear or anxiety had higher levels of worry about radiation than those without intense fear or anxiety throughout the study period. Meanwhile, among those who experienced direct damage in the Great East Japan Earthquake, although their worry about radiation was greater than that of those who did not experience direct damage at Times 1 and 2, it reduced faster with time, and their 95% confidence intervals overlapped after Time 3 (i.e., 32 months after the baseline survey). Disaster-related family problems did not significantly interact with time and maintained their effects on worry about radiation throughout the study period. Concerning the evacuation to avoid radiation exposure after the nuclear power plant accident, while worry about radiation of those who evacuated was higher at the baseline survey, it declined sharply and became lower than that of those who did not evacuate after Time 4 (i.e., 44 months after the baseline survey), although their 95% confidence intervals overlapped.

## 4. Discussion

This longitudinal study of non-evacuee community residents in Fukushima Prefecture conducted five to ten years after the nuclear power plant accident examined the duration of the effects of risk factors on worry about radiation. Concerning disaster-related experiences, the effects of intense fear or anxiety immediately after the nuclear power plant accident persisted 10 years after the accident, although they had weakened with time. The effects of disaster-related family problems had persisted without interaction with time. The effects of direct damage and evacuation experience diminished with time. Concerning the sociodemographic characteristics, being under the age of 65, having low educational attainment, and being married were associated with greater worry about radiation; however, the association with age weakened with time.

While worry about radiation among those who experienced direct damage in the Great East Japan Earthquake was higher than that among those who did not at the baseline survey, the difference diminished with time and became non-significant at the third survey, about seven years after the accident. The direct damage experienced during the disaster, which was a risk factor of worry about radiation in the early phase of the nuclear power plant accident [5,9,12], may not contribute to the persistence of worry about radiation in the longer term.

Although worry about radiation among those who felt intense fear or anxiety immediately after the nuclear power plant accident declined faster, it was continuously higher than among those who did not feel such intense fear or anxiety until 10 years after the accident. Intense fear or anxiety immediately after the accident may indicate the existence of traumatic events more directly than experiences of direct damage. The finding that greater levels of worry about radiation among those who felt intense fear or anxiety at the time of the accident persisted for a decade, as well as a previous study’s finding that worry about radiation predicted the presence of posttraumatic stress symptoms seven years after the accident [6], may suggest the long-lasting effects of traumatic events.

Disaster-related family problems significantly affected worry about radiation and their effects did not attenuate with time. These problems, consisting of the deterioration of family relationships and family separation, may lead to chronic stress exerting persistent effects over time. A previous study reported that high social support is associated with recovery from posttraumatic stress disorder after a disaster [19]. Given the similarity of worry about radiation with trauma-related disorders in terms of its association with traumatic events and posttraumatic stress symptoms [6], it is possible that the decrease in social support from family members stemming from disaster-related family problems had deleterious effects on worry about radiation.

Regarding the evacuation after the accident, those who evacuated displayed higher worry about radiation than those who did not at the baseline survey. However, their worry declined faster and became lower, although not significantly, after the fourth survey, about nine years after the accident. Individuals with greater worry about radiation were more likely to evacuate the site after the accident to avoid further radiation exposure. This may explain their higher worry about radiation in the early phase of the accident. Simultaneously, their avoidance of further radiation exposure could contribute to the fast decline in their subsequent worry about radiation. A previous study showed that those who had evacuated voluntarily had higher radiation health anxiety than those who did not evacuate nine years after the Fukushima nuclear power plant accident [20]; these findings are inconsistent with our results. Our study included those who evacuated only their family but not themselves in the evacuated group, while the previous study excluded such people, which might explain the inconsistency to some extent. Wide confidence intervals of worry about radiation in our evacuated group may indicate that it contained a wide range of people. From another perspective, other studies reported that mastery, a psychological resource indicating a perception of control over one’s life, contributed to the recovery from stress-related disorders among those who experienced traumatic events [21,22]. Our results of a faster decline in worry about radiation among those who evacuated might suggest that they were able to retain a sense of control over their lives because of the evacuation.

Concerning sociodemographic characteristics, individuals who were younger than 65, had low educational attainment, and were married had greater worry about radiation. The level of education may be associated with the understanding of the risk of radiation; thus, those with poor understanding at the early phase of the accident may tend to retain unnecessary worry about radiation. Household income was not significantly associated with worry about radiation, which was a finding that differed from a previous study [5]. Although income is an indicator of socioeconomic status and is believed to be strongly associated with education, understanding the risk of radiation may be more directly related to education. Individuals who were married had greater worry about radiation compared to those who were not because they might have additional concerns for the health of their family members and thus greater worry about radiation. Annual surveys of evacuees after the Fukushima nuclear power plant accident consistently reported that more respondents were concerned about the possibility of genetic effects of radiation exposure (i.e., health effects on their future children and grandchildren), compared to the possibility of delayed effects (i.e., damage to their own health in later life) [9,14]. Worry for the future generation may persist for a long time. The effects of age diminished with time, which may partly be due to the aging of the study participants since the accident.

Our study has several limitations. First, the response rate of the baseline survey was not very high (41.6%), which might have led to a selection bias. Individuals without disaster-related experiences or worry about radiation may have been less likely to participate in this survey due to their lack of interest, by which the associations observed between disaster-related experiences and worry about radiation might have been underestimated. Second, the response rate of the following surveys decreased over time. If those without worry about radiation were more likely to drop out, the estimation of worry about radiation based on the survey participants may have been overestimated with the passage of time. However, we confirmed that in terms of the level of worry about radiation, there were no statistically significant differences between those who responded to the subsequent survey and those who did not, at least in the preceding survey. We also found that the levels of worry about radiation in each survey were almost the same when calculated only among those who responded to all five time surveys. Therefore, we think the possibility of gradual overestimation with time due to the dropout is minimal. Third, although we revealed that the effects of disaster-related experiences on worry about radiation, except for family problems, had weakened during the five to ten years after the nuclear power plant accident, the relationships of time and these effects were not explored during the first five years after the accident. It may be possible that their relationships were not the same in the early phase of the disaster.

## 5. Conclusions

We examined the duration of the effects of risk factors on worry about radiation using data obtained from a longitudinal questionnaire survey following a random sample of non-evacuee community residents five to ten years after the Fukushima nuclear power plant accident. Individuals who experienced intense fear or anxiety after the nuclear power plant accident or family problems stemming from the disaster may continue to experience worry about radiation for more than 10 years, and, thus, need careful long-term monitoring after nuclear power plant accidents.

Regarding long-term monitoring after nuclear power plant accidents, healthcare professionals providing direct care for the affected residents, such as public health nurses, are required to consider the persistent worry about radiation even if the residents do not mention these concerns. For policymakers planning continuous support, it is necessary to prepare the system for dealing with worry about radiation in case it re-emerges in the future.

## Figures and Tables

**Figure 1 ijerph-19-16943-f001:**
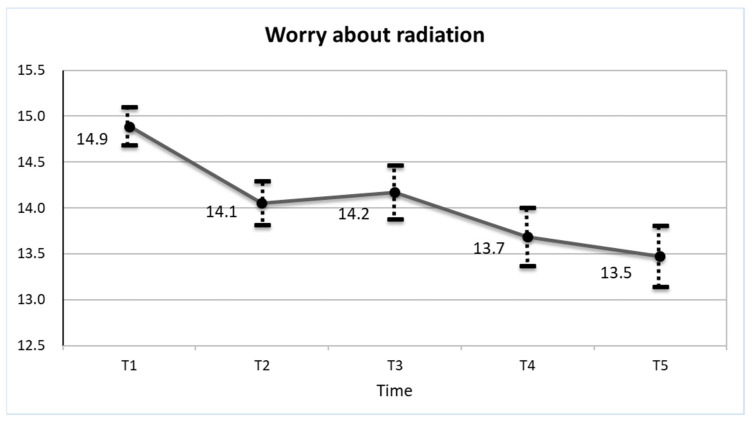
Time course change of worry about radiation (means and 95% confidence intervals) five to ten years after the nuclear power plant accident. T1: baseline survey or five years after the nuclear power plant accident (*n* = 1754); T2: second survey or 20 months after T1 (*n* = 1215); T3: third survey or 32 months after T1 (*n* = 896); T4: fourth survey or 44 months after T1 (*n* = 770); T5: fifth survey or 56 months after T1 (*n* = 701).

**Figure 2 ijerph-19-16943-f002:**
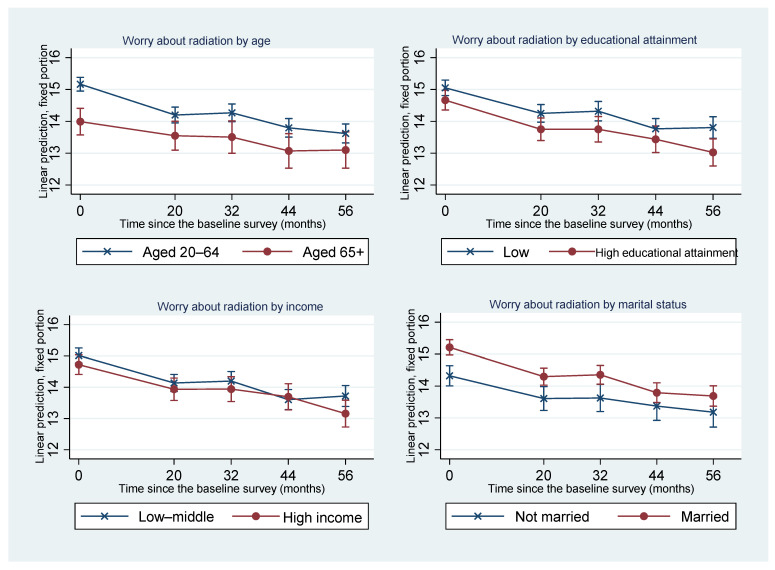
Changes in the adjusted predicted means of worry about radiation and 95% confidence intervals of those with and without each risk factor.

**Figure 3 ijerph-19-16943-f003:**
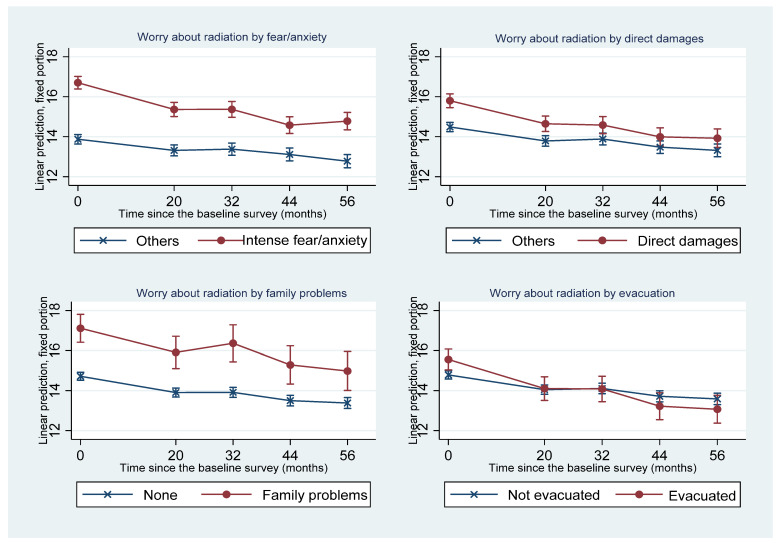
Changes of the adjusted predicted means of worry about radiation and 95% confidence intervals of those with and without each disaster-related experience.

**Table 1 ijerph-19-16943-t001:** Characteristics of the participants in the baseline survey conducted five years after the Fukushima nuclear power plant accident. (*n* = 1825).

				*n*	%/SD
Sociodemographic characteristics					
	**Age, years**				
		20–39		804	44.1
		40–64		612	33.5
		65+		409	22.4
		Mean (Standard Deviation)		47.1	(17.3)
	**Sex**				
		Men		859	47.1
		Women		966	52.9
	**Educational attainment**				
		Up to high school		1127	61.8
		Junior or technical college or higher		698	38.3
	**Annual household income**				
		Low (Less than JPY 250 million)		430	23.6
		Middle (JPY 250–500 million)		724	39.7
		High (JPY 500 million or more)		671	36.8
	**Marital status**				
		Married		1159	63.5
		Separated, divorced, bereaved, unmarried, or unknown		666	36.5
	**Residential area**				
		Eastern coastal area (Hama-dori)		138	7.6
		Central area (Naka-dori)		1009	55.3
		Western mountainous area (Aizu)		678	37.2
Disaster-related experiences					
	**Fear or anxiety immediately after the accident**				
		None to moderate		1166	63.9
		Intense		659	36.1
	**Direct damage (ref. none)**				
		Injured		14	0.8
		Family member injured or dead		79	4.3
		Loss of job or temporary absence from work		260	14.3
		Loss of house or property		359	19.7
		Cumulative damage			
			0	1252	68.6
			1	449	24.6
			2+	124	6.8
	**Disaster-related family problems (ref. none)**				
		Deterioration of family relationships		68	3.7
		Family separation		96	5.3
		Cumulative damage			
			0	1681	92.1
			1+	144	7.9
	**Evacuation to avoid radiation exposure**				
		Not evacuated		1556	85.3
		Evacuated		269	14.7

SD, standard deviation; ref., reference.

**Table 2 ijerph-19-16943-t002:** The effects of time, sociodemographic characteristics, and disaster-related experiences on worry about radiation from five years after the Fukushima nuclear power plant accident. (N = 1825).

			Bivariate Associations			Model 1			Model 2		
			Coef.	SE	*p*	AdjustedCoef.	SE	*p*	AdjustedCoef.	SE	*p*
Time (month)			−0.03	0.00	<0.001	−0.03	0.00	<0.001	−0.01	0.00	0.027
**Sociodemographic characteristics**											
	Age (65+)		−0.81	0.22	<0.001	−0.88	0.21	<0.001	−1.10	0.23	<0.001
	Sex (women)		−0.11	0.19	0.554	−0.35	0.17	0.034	−0.35	0.17	0.032
	Educational attainment (junior or technical college or higher)		−0.30	0.19	0.115	−0.47	0.18	0.009	−0.41	0.20	0.034
	Household income (high)		−0.12	0.19	0.534	−0.25	0.18	0.151	−0.30	0.19	0.127
	Marital status (married)		0.68	0.19	<0.001	0.74	0.18	<0.001	0.87	0.20	<0.001
	Residential area (eastern coastal area or central area)		1.91	0.19	<0.001	1.27	0.18	<0.001	1.27	0.18	<0.001
**Disaster-related experiences (ref. none)**											
	Fear or anxiety immediately after the accident (intense)		2.74	0.18	<0.001	2.30	0.18	<0.001	2.60	0.19	<0.001
	Direct damage		1.88	0.19	0.001	0.97	0.19	<0.001	1.18	0.21	<0.001
	Disaster-related family problems		3.41	0.34	<0.001	2.18	0.33	<0.001	2.09	0.36	<0.001
	Evacuation to avoid radiation		1.96	0.26	<0.001	0.25	0.26	0.337	0.61	0.28	0.030
**Interaction of the time and sociodemographic characteristics**											
	Time^×^Age (65+)								0.01	0.01	0.041
	Time^×^Educational attainment (junior or technical college or higher)								0.00	0.00	0.606
	Time^×^Household income (high)								0.00	0.00	0.589
	Time^×^Marital status (married)								−0.01	0.00	0.110
**Interaction of the time and disaster-related experiences**											
	Time^×^Fear or anxiety immediately after the accident (intense)								−0.02	0.00	0.001
	Time^×^Direct damage								−0.01	0.00	0.014
	Time^×^Disaster-related family problems								0.00	0.01	0.570
	Time^×^Evacuation to avoid radiation exposure								−0.02	0.01	0.002

Coef., coefficient; SE, standard error; ref., reference.

## Data Availability

The datasets used and analyzed during the current study are available from the corresponding author upon reasonable request.

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
