# Peer review of "Worry about Radiation and Its Risk Factors Five to Ten Years after the Fukushima Nuclear Power Plant Disaster"

_ijerph, 2022, doi:10.3390/ijerph192416943_

Round 1
Reviewer 1 Report
In this manuscript, the authors evaluated the worry about radiation after the Fukushima Nuclear Power plant accidents and how it changes with time. The authors conducted questionnaire surveys and performed the statistical analysis. The analysis identified risk factors related to the worry about radiation.
The authors have collected valuable data in understanding important research questions, how fear and anxiety change after an accident, and what the associated risk factors are.
In general, the materials and methods are well-written. However, it needs to be made clear how the surveys are collected. Is collected by phone call, by mail, or in person? I would also want to know whether the response rate is correlated to residential areas. It would also be helpful if the authors could provide a sample questionnaire in the supplemental material.
In Figure 1, why the number of samples at T1 is 1,754 rather than 1,825? What does the dot located at the center of the figure mean?
It was not clear to me why the authors made specific choices for the analysis. Why did the authors choose 65 as the threshold for age? How do the models look if 40 is chosen as the threshold? Will the result be significant if we use the low household income versus middle and high income as the variable?
I was unable to understand Table 2. Why is Model 1 associated with two sets of coefficients? What are the differences between the two sets?
In Model 2, the authors obtained statistically significant results for the interaction of time and other variables. However, I did not find explicit discussion related to the results. Are there implications associated with the significance of the interaction terms?
The authors have sufficiently discussed the limitations of the manuscript. However, I suggest the authors check whether the subsequent dropout correlates with the worry score in the previous time point to detect the potential bias of the analysis.
Reviewer 2 Report
Dear authors,
Thank you for the opportunity to review your manuscript. This is an interesting and important study to support people who have suffered the ill-effects of nuclear power plant disasters. As the manuscript is well-written, I will only make some minor comments.
Line 229
According to the authors, “The difference diminished quickly.” I wonder if it is appropriate to say “quickly” because 32 months from the baseline survey is not a short period. Furthermore, the baseline survey was conducted five years after the disasters, and this result has come 92 months after the disasters. Please consider revising this word.
Conclusions
Mentioning additional implications for professionals or policymakers would increase the value of this paper. What should professionals such as public health nurses do as interventions? How should policymakers draft rules to support people who have suffered the consequences of a disaster?
Figure 1
Is the overall decreasing score significantly meaningful? Please consider conducting a statistical analysis such as repeated ANOVA. If the difference is not significant, this figure will produce misleading results. Including the standard error bar aids in the interpretation of the figure. Is the dot near T3 an error? Please verify.
Reviewer 3 Report
INTRODUCTION
Fukushima nuclear power plant accident, greater worry about radiation was found to be associated with young age, having low educational attainment, low income, being married, and disaster-related experiences (PLEASE EXPLAIN IN DETAIL THE ASSOCIATION WITH Worry with age, low educational attainment, low-income dan married status.
2. Materials and Methods
Please explain. Is there the same questionnaire given to respondent during 5 years
The baseline survey was conducted between February and April 2016. The 84 second, third, fourth, and fifth surveys were conducted in 20, 32, 44, and 56 months (Please justify why the second to the fifth survey was conducted in different months.
Explain - the number of questionnaires distributed dan return rate from the participants
2.2.3. Disaster-related Experiences
Explain - is this questionnaire adapted or adopted
Round 2
Reviewer 1 Report
Thanks for the response. The authors have addressed most of my concerns. There is still a dot at the center of Figure 1.
Author Response
I deleted the old Figure 1 (which was still in the paper because I used track changes function) and added new one below (to which 95% CIs were added). I checked my file and I found a dot only in the old Figure 1. I have attached the file with removing the track changes. Please check the new Figure 1 in this file.
